# Iridoids: Research Advances in Their Phytochemistry, Biological Activities, and Pharmacokinetics

**DOI:** 10.3390/molecules25020287

**Published:** 2020-01-10

**Authors:** Congcong Wang, Xue Gong, Agula Bo, Lei Zhang, Mingxu Zhang, Erhuan Zang, Chunhong Zhang, Minhui Li

**Affiliations:** 1Baotou Medical College, Baotou 014060, Inner Mongolia, China; WangCongCong0@163.com (C.W.); gongxue_2017@yeah.net (X.G.); agula372000@126.com (A.B.); zhang_mingxu@yeah.net (M.Z.); huan15733941992@163.com (E.Z.); 2Faculty of Pharmacy, Inner Mongolia Medical University, Hohhot 010110, Inner Mongolia, China; 15771378662@163.com; 3Inner Mongolia Key Laboratory of Traditional Chinese Medicine Resources, Baotou Medical College, Baotou 014060, Inner Mongolia, China; 4Inner Mongolia Institute of Traditional Chinese Medicine, Hohhot 010020, Inner Mongolia, China

**Keywords:** iridoids, phytochemistry, biological activities, pharmacokinetics

## Abstract

Iridoids are a class of active compounds that widely exist in the plant kingdom. In recent years, with advances in phytochemical research, many compounds with novel structure and outstanding activity have been identified. Iridoid compounds have been confirmed to mainly exist as the prototype and aglycone and Ι and II metabolites, by biological transformation. These metabolites have been shown to have neuroprotective, hepatoprotective, anti-inflammatory, antitumor, hypoglycemic, and hypolipidemic activities. This review summarizes the new structures and activities of iridoids identified locally and globally, and explains their pharmacokinetics from the aspects of absorption, distribution, metabolism, and excretion according to the differences in their structures, thus providing a theoretical basis for further rational development and utilization of iridoids and their metabolites.

## 1. Introduction

Iridoids, a large and still expanding class of cyclopentane pyran monoterpenes, are composed of two basic carbon frameworks, substituted iridoids, and secoiridoids. They are more prevalent in the kingdom Plantae, especially in the dicotyledonous plants from Scrophulariaceae, Pyrolaceae, Oleaceae, Labiatae, Rubiaceae, and Gentianaceae [1,2]. Based on the structure, these compounds can be divided into four groups: iridoid glycosides, secoiridoid glycosides, non-glycosidic iridoids, and bis-iridoids. Iridoids have hemiacetal hydroxyl groups and are active in nature. Besides, they are mostly in the form of glycosides and are combined with glucose at the C-1 hydroxyl group [3,4]. Most glycosides share similar properties such as strong modifiability and rapid absorption; hence, they are popular sources of lead compounds. Their prominent effects in various diseases have shown their importance in pharmaceutical chemistry research. As one of the active components used in natural medicine and traditional Chinese medicine, iridoid compounds have many biological effects such as liver protection, anti-inflammatory and anti-tumor effects [5,6,7]. However, in recent years, there has been no systematic arrangement and summary of iridoids. This review paper aims to characterize iridoids based on their phytochemistry, biological activities, and pharmacokinetics, through the analysis and comparative study of the structure and function of known iridoids. The characterization will provide a framework for new medicine development and research, enabling the exploration of the remarkable medicinal potential of these compounds.

The systematic summary in this paper was produced by searching for relevant information about iridoids on websites such as Google Scholar, PubMed, CNKI, Baidu Scholar, SciFinder Scholar, TPL (www.theplantlist.org), and Web of Science. The following words and/or phrases were used in various combinations, “iridoids”, “iridoids phytochemistry”, “iridoids classification”, “iridoid glycosides”, “secoiridoid glycosides”, “bis-iridoids”, “non-glycosidic iridoids”, “iridoids activities”, “iridoids pharmacological”, “iridoids pharmacokinetic”. More than 100 scientific works of literature were consulted from 2009 to 2019.

## 2. Phytochemistry

Iridoids belong to monoterpenoids, which are acetal derivatives of iridodial [8]. Because of the unstable nature of its C_1_-OH group, iridoids often react with sugar to form glycosides. According to the integrity of the cyclopentane unit, it can be divided into two types: iridoid glycosides (including geniposide, loganin, acetylbarlerin, deacetylasperulosidic acid, and brasoside) and secoiridoid glycosides (including swertiamarine, gentiopicroside, qinjiaoside A, qinjiaoside B, sweroside, and oleuropein). In addition, the dimer of iridoids called as bis-iridoids (including cantleyoside, laciniatoside I-II, and sylvestroside I) and unglycosylated iridoids called as non-glycosidic iridoids (including acevaltrate and valtrate) [9,10] are also found in nature. Compounds with such structures possess a variety of physiological activities and are used as sedating and antipyretic agents [8,11]. Tundis et al. have detailed summaries of active compounds prior to 2008 [12], Table 1 lists the active iridoids from 2009 to 2019 in the last 11 years. Three basic skeletons of iridoids are elucidated in Figure 1.

### 2.1. Iridoid Glycosides

The iridoid glycosides C-1 are polyhydroxyl linked and polyglycoside formed, and most of them are β-d-glucoside. Among the isolated iridoid glycosides, some rare new structures have been found, such as asperuloside, which has a ketone functional group at C-6. This cyclopentane ring with ketoyl function, especially at C-6, is a rare compound in the plant kingdom [45]. Besides, Iridoid glycosides with apiofuranosyl moieties have been reported for the first time in *Sambucus williamsii* Hance. Based on the extensive spectroscopic analysis (NMR and HRESIMS) the structures were elucidated as williamsoside A and B [46].

Furthermore, 6-*O*-α-l-(2″-acetyl-4″-*O*-*trans*-isoferuloyl) rhamnopyranosyl catalpol is acylated and esterified at the C-2 and C-4 positions such compounds have never been separated from natural resources so far [26]. Vestena et al. isolated and purified brasoside from both *Verbena litoralis* and *Verbena montevidensis* [27]. In addition, asperuloside with a rare iridoid structure was isolated from the root and stem of *Ronabea emetica* [30]. Their structures are presented in Figure 2.

### 2.2. Secoiridoids Glycosides

Secoiridoids have many structural differences compared with the iridoid glycosides [47]. Compound **42**–**57** are active compounds from the secoiridoid class, which are widely distributed in Gentianaceae, Nymphaeaceae, Caprifoliaceae, and Oleaceae, and other plants, and are more common in Gentianaceae and Swertia.

The C_7_–C_8_ in secoiridoids in the parent nucleus of these compounds is usually broken to form a cleavage ring. The C-7 and C-11 compounds sometimes form a 6-membered lactone structure after ring splitting. Secoiridoid rearrangements are relatively rare, and nuezhenelenoliciside is one such compound, which undergoes a break in the chemical bond between C-1 and O-2 and generates a new secoiridoid ring between C-8 and O-2. The effect of nuezhenelenoliciside on the proliferation of pre-osteoblast cells has been reported [40]. Pagide is considered the second member of the secoiridoids family of rare, naturally occurring organisms that are produced through an iridoid biosynthesis pathway based on their unique C-4 acyl substitution and C-8 glycosylation [48]. Their structures are presented in Figure 3.

### 2.3. Bis-Iridoids

In addition to the compounds, there exists a class of bis-iridoids in nature that are dimeric iridoids formed from the structures of two iridoids [9]. Cantleyoside, laciniatoside I-II, sylvestroside I, sylvestroside III-IV, sylvestroside III dimethyl acetal, and sylvestroside IV dimethyl acetal are mainly bis-iridoids isolated from *Pterocephalus hookeri* (C.B. Clarke) Höeck, which have been shown to have anti-inflammatory and analgesic activities [41]. Sclerochitonoside C has a similar structure, except that the C-7 position of the iridoid is directly connected to the C-4 position of the secoiridoid, which comprises subunits based on the 4′-hydroxyphenylethyl esters of 8-epiloganic acid and its secoiridoid analogue [43].

Furthermore, structures similar to those of laxoside have not been previously described in international catalogues. Such a structure is formed by acid hydrolysis to produce d-glucose and d-galactose; alkali hydrolysis then produces *trans*-*p*-methoxycinnamic acid and a slightly polar glycoside. Spectral analysis data shows that the structure of 6‴-*O*-[6-*O*-(4″-*trans*-*p*-methoxycinnamoyl)-5-hydroxyaucubigenin-(1→1′)-*O*-β-d-galactopyranosyl]-6″″-*O*-*trans*-p-methoxycinnamoylaucubin uses glucose as a bridge to link the rare iridoid dimers [49]. Their structures are presented in Figure 4.

### 2.4. Non-Glycosidic Iridoids

A few iridoid compounds, called non-glycosidic iridoids, are also found in nature. Mueller et al. isolated valtrate, 1-β-acevaltrate, and acevaltrate from *Momordica charantia* L.; these compounds did not contain glycosides, and **70** and **71** were isomers [44]. In addition, two non-glycosidic iridoids, **72** and **73**, were isolated from eudodotes *Gardenia jasminoides* Ellis [19]. Their structures are presented in Figure 5.

## 3. Biological Activities

Now, more and more iridoids have been discovered, and their pharmacological effects are further studied, such as neuroprotective, hepatoprotective, anti-inflammatory, hypoglycemic, hypolipidemic, and antitumor activities. Their main pharmacological activities and the underlying mechanisms are shown in Figure 6.

### 3.1. Neuroprotective Effects

Regeneration and repairing of the central nervous system (CNS) is a complex process with multiple links and factors [11]. Nerve growth factors (NGF) are involved in many functions of the nervous system, such as growth, survival, and regeneration, in addition to repairing the neurons from various wounds. However, they cannot penetrate the blood-brain barrier and are easily hydrolyzed by hydrolases [50,51]. Therefore, researchers are trying to find natural lipophile compounds with the properties of endogenous neurotrophic factors to induce neuronal differentiation and regeneration. To date, all the available data suggest that the iridoids are a class of natural lipophile compounds with the properties of endogenous neurotrophic factors, which could be considered as potential leads for the treatment of neurodegenerative diseases [52]. The protective effects of loganin against PD mimetic toxin 1-methyl-4-phenylpyridinium (MPP+) and the important roles of insulin-like growth factor 1 receptor (IGF-1R) and glucagon-like peptide 1 receptor (GLP-1R) in the neuroprotective mechanisms of loganin have been demonstrated. It has been shown that in primary mesencephalic neuronal cultures treated with or without MPP+, loganin protects against MPP+ -induced apoptosis by up-regulating the expression of anti-apoptotic proteins and pro-apoptotic proteins, and also enhances the expression of neurotrophic signals by up-regulating the expressions of IGF-1R, GLP-1R, p-Akt, BDNF, and tyrosine hydroxylase. In addition, it can reduce the production of MPP+-induced reactive oxygen species (ROS), up-regulate GAP43, and down-regulate membrane-rhoA/ROCK2/p-LIMK/p-cofilin to reduce MPP+ -induced neuron damage. Furthermore, the presence of IGF-1R antagonists (AG1024) or GLP-1R antagonists (exendin 9-39) weakened the protective effect of loganin on MPP+ -induced cytotoxicity and apoptosis, and decreased neuron length and ROS production. A series of experiments showed that loganin could enhance neurotrophic signals, activate IGF-1R/GLP- 1R, inhibit RhoA/ROCK pathway, and alleviate MPP+ induced apoptosis and death, neuron damage, oxidative stress, and other mechanisms to achieve neuroprotective effect [53].

Studies have found that abnormal tau protein hyperphosphorylation could be one of the main causes of Alzheimer’s disease (AD). The concentration of Protein phosphatase 2A (PP2A) affects the tau protein hyperphosphorylation, which in turn is concentration-dependent. To investigate the effect of cornus iridoid glycoside (CIG) on enhancing the activity of PP2A in Cornus officinalis, Sk-N-SH cells were exposed to 20 nmol/L okadaic acid (OA, PP2A inhibitor) for 6 h to induce tau hyperphosphorylation to determine the effect of CIG on PP2A activity and on the post-translational modification of the PP2A catalytic subunit C (PP2Ac). The results showed that OA increased PP2Ac phosphorylation and tau hyperphosphorylation, while CIG preincubation significantly reduced OA-induced tau hyperphosphorylation at Ser 199/202 and Ser 396, inhibited PP2Ac phosphorylation at Tyr 307, increased Src phosphorylation, and restored PP2A activity. In conclusion, CIG can inhibit tau hyperphosphorylation by activating PP2A, reducing the p-SRC level and PP2Ac phosphorylation at Tyr307 [54]. Another compound, corn furfural B isolated for the first time from the ethanol extract of dogwood fruit, has been shown to possess neuroprotective activity by reducing the corticosterone-induced neuronal damage in PC12 cells [55]. However, the neural network is a complex system, and iridoids are a large family, with different compounds having different effects on the nervous system. Especially, the compounds with large differences in their chemical structure need to be studied in more detail. Hence, there are limitations in the study of pharmacological effects and their mechanisms.

### 3.2. Hepatoprotective Effects

Many studies have shown that the current effective treatments for liver disease are often associated with side effects [56]. It is highly prevalent in the current scenarios to explore the therapeutic value of natural medicinal preparations or to explore their novel effects. The hepatoprotective effects of swerodesanthin and swerodesanthin from *Swertia mussotii* Franch were evaluated by using the model of α-naphthol isocyanate (ANIT)-induced liver injury. After oral administration of swertiamarin (20 mg/kg) or swertianolin (20 mg/kg) for seven days, alanine transaminase, aspartate transaminase, and total direct bilirubin levels in mice were significantly reduced at α-naphthyl isothiocyanate-induced levels, while, bile flow was significantly increased (*p* < 0.01). The study also showed that in the cholestatic hepatitis model, the inducer ANIT could promote bile secretion; reduce AST, ALT, ALP enzyme activity; and reduce TBIL and DBIL levels which indicates the role of cycliterpene in the treatment of cholestatic hepatitis [57].

Another study tested the vegetable oil- and carbon tetrachloride-induced liver fibrosis rat models to evaluate and characterize the hepatoprotective effects of iridoid glycosides from *Boschniakia rossica*. The rats were continuously treated with iridoid glycosides (200 mg/kg) for 10 days and then their blood and liver tissues were harvested. When compared to the normal control group, serum ALT, AST, and TBIL levels were significantly increased in the model group (*p* < 0.05). HE staining and Masson staining showed that the hepatocytes in the model group exhibited degeneration, necrosis, and hepatic fibrosis. Western blot analysis showed that the expression of α-SMA was significantly increased in the liver tissue from the model control group (*p* < 0.05). The rats from *Boschniakia rossica* iridoid glucoside group showed significantly low levels of serum ALT, AST, and TBIL (*p* < 0.05), with markedly reduced liver fibrosis compared with that in the model control group. The expression of α-SMA in *Boschniakia rossica* iridoid glucoside group was significantly lower than that in the model control group (*p* < 0.05). In essence, *Boschniakia rossica* iridoid glycoside could inhibit liver fibrosis by inhibiting the activation of hepatic stellate cells, which are shown to exhibit some protective effects against liver damage [58].

To determine the protective effect of CIG on hepatocyte damage induced by d-galactosamine (GalN) combined with TNF-α, L-02 human liver cells were cultured in vitro to establish a D-GalN/TNF-α-induced damage model in L-02 cells. The total antioxidant capacity (T-AOC) of superoxide dismutase (SOD), malondialdehyde (MDA) and intracellular calcium concentration were taken as the index, and 44 mg/L D-GalN combined with 100 μg/L TNF-α could be used as the dose in the cell damage model. The pre-protection group with high, medium, and low CIG doses (10, 20, 100 mg/L) significantly increased the activity of damaged cells and SOD activated T-AOC capacity of damaged cells, which was significantly different from that in the model group (*p* < 0.05). The activity of MDA might also have been reduced, but the decrease of MDA activity in the low concentration group was not obvious. Furthermore, western blotting showed that CIG at different concentrations significantly reduced the expression of endoplasmic reticulum stress-related protein PERK, EIF-2α, and apoptosis-associated protein caspase-3. This suggests that the mechanism may be related to the enhancement of the antioxidant capacity of cells, the reduction of damage caused by endoplasmic reticulum stress, and the reduction of the expression of apoptosis-related proteins [59]. At present, the related research on the hepatoprotective activity of iridoid compounds focuses on anti-inflammatory factors and lipid peroxide [60,61]. Some studies have shown that iridoid glycosides can also show anti-inflammatory and anti-fibrosis activity by decreasing the expression of TGF-β1 protein [62,63]. In addition, some studies have shown that promoting the opening of MPTP, resulting in the release of apoptosis factors, inhibiting the oxidation of mitochondria and respiratory chain function, making the electron transport disorder in mitochondria, interfering with ATP synthesis is also an important mechanism of liver protection [64,65]. However, the hepatoprotective activity of iridoid is less studied in the aspect of mitochondrial damage, which should be paid attention to in the future.

### 3.3. Anti-Inflammatory Activities

In the search for anti-inflammatory compounds, special interest has been directed towards iridoids [66,67,68,69,70,71]. The anti-inflammatory and analgesic activities of some diterpene components of pterygium pterygii were evaluated. The analgesic effect of *Polygonum cuspidatum* was studied by the hot plate method and acetic acid twisting method. Compared with the control group, BCPH (50, 100 mg/kg) reduced twisting times (17.25 ± 6.24 and 21.25 ± 3.65) and had a significant analgesic effect on the acetic acid-induced pain (*p* < 0.01). In the hot plate latent trial, the efficacy of a high dose of BCPH (100 mg/kg) was due to the control medicine Rotundine (20 mg/kg). The anti-inflammatory effects on carrageen-induced posterior foot swelling in rats and dimethylbenzene induced ear swelling in mice were observed. It was found that BCPH at 100 mg/kg had a significant inhibitory effect on ear edema induced by dimethylbenzene in mice. Moreover, BCPH inhibited TNF-α-induced NF-κB-dependent promoter activity in a dose-dependent manner. These results indicated that Polygonum diocyclic ether can significantly reduce the pain caused by acetic acid and reduce ear swelling in mice. At the cellular level, bis-iridoid aglycones reduced TNF-α- and LPS-induced NF-κB activation and reduced inflammatory factors (inhibiting the production of free radicals) [41].

Loganin, an iridoid glycoside present in several herbs, including *Flos lonicerae*, *Cornus mas* L, and *Strychnos nuxvomica*, is a valuable medication with anti-inflammatory effects [72]. To characterize the roles of loganin in osteoarthritis and its specific signaling pathway, chondrocytes were administrated with IL-1ss and supplemented with or without LY294002 (a classic PI3K/Akt inhibitor). The apoptotic level, catabolic factors (MMP-3 and MMP-13 and ADAMTS-4 and ADAMTS-5), extracellular matrix (ECM) degradation, and activation of the PI3K/Akt pathway were evaluated using western blotting, PCR, and an immunofluorescent assay. The degenerative condition of the cartilage was evaluated in vivo using the Safranin O assay. The expression of cleaved-caspase-3 (C-caspase-3) was measured using immunochemistry. The results suggested that loganin suppressed the apoptosis, reduced the release of catabolic enzymes, and decreased the ECM degradation of IL-1ss-induced chondrocytes. However, suppressing PI3K/Akt signaling using LY294002, alleviated the therapeutic effects of loganin in chondrocytes. Loganin partially attenuated cartilage degradation while inhibiting the apoptotic level. This work revealed that loganin treatment attenuated IL-1ss-treated apoptosis and ECM catabolism in rat chondrocytes via regulation of the PI3K/Akt signaling, revealing that loganin is a potentially useful treatment for osteoarthritis [73]. In addition, isolated camptoside from *Camptosorus sibiricus* Rupr. (Aspleniaceae) which exhibited inhibitions of nitric oxide production in lipopolysaccharide-induced RAW 264.7 macrophages with IC50 values of 11.2 Μm [74]. Apart from the above research, the future is it necessary to inflammation related to signal transduction pathways, AA metabolic pathways of ring oxidase (COX), isomerase and its downstream of prostaglandin (PGs), synthetase system and corresponding receptors and phosphodiesterase isoenzyme more anti-inflammatory pathway and links such as anti-inflammatory effect of iridoid molecular mechanism for more in-depth research.

### 3.4. Hypoglycemic and Hypolipidemic Activities

Type 2 diabetes is a common, lifestyle-related chronic disease, and is often accompanied by various complications. The pathogenesis of type 2 diabetes is insulin resistance and dysfunction of islet β cell. Hyperinsulinemia and hyperlipidemia are important parts of insulin resistance syndrome and important risk factors of cardiovascular disease [75]. In one pharmacological investigation, it has been found that the compound of iridoids can effectively control the hyperglycemia and hyperlipidemia. The effects of CIG were studied in a mouse model of diabetes induced by a high-fat diet (HFD) and streptozotocin (STZ). The results showed that CIG (IG: 75, 150, 300 mg/kg) could significantly improve the glucose tolerance of diabetic mice, and CIG could enhance the expression of PI3K-Akt/PKB pathway-related proteins in insulin metabolism, through which the hyperglycemia and hyperlipidemia of diabetic mice induced by HFD and STZ could be improved [76].

By studying the glucose-sensing effect of Glucocorticoid (POMC) neurons genetically expressed in mutant immunoglobulin receptor 6.2 (Kir6.2) subunit (encoded by KCNJ11 gene), researchers found that this mutation can prevent ATP mediated ATP sensitive potassium pathway (KATP) closure and increase the whole-body glucose load of obese mice fed with a high-fat diet. This indicates that the loss of glucose sensing like effect of POMC neurons influences the formation of type 2 diabetes. As a membrane-permeable molecule, genipin (20 μM) can block the proton leakage mediated by uncoupling protein (UCP2), and UCP2 can negatively regulate the glucose-sensing effect of POMC neurons, resulting in the loss of glucose sensing effect. When increasing the culture of islet β cells, genipin can increase the potential of mitochondrial membrane, thus increasing the ATP level and closing the KATP pathway, stimulating glucose excitatory neurons and stimulating insulin secretion. This result showed that genipin has a potential therapeutic effect on diabetes [77].

A model of diabetes mellitus in rabbits was established by intravenous injection of alloxan at a dose of 150 mg/kg. After diabetes induction, the plasma, erythrocyte malondialdehyde (MDA) and blood glucose of all diabetic animals increased significantly. For the next 16 weeks, the treatment group took 20 mg/kg OLE orally every day. At the beginning of the treatment, compared with that in diabetic control, the blood glucose levels in rabbits increased continuously throughout the study period. The blood glucose level of the treated rabbits, however, decreased significantly in the eighth week. The results showed that the activity of superoxide dismutase (SOD), the level of MDA, and blood glucose were significantly increased, indicating that oleuropein can be used as a hypoglycemic and antioxidant agent to prevent complications and oxidative stress caused by diabetes [78]. At present, the mechanism of hypoglycemia mainly includes insulin stimulation and repair of damaged pancreas; effects on glucose metabolism; increases insulin sensitivity, improves insulin resistance, and regulates intestinal flora [79,80,81,82,83]. Iridoids to regulate intestinal flora’s research are also insufficiently thorough, the active ingredient from iridoid monomers adjusts the perspective of intestinal flora, by adjusting the proportion of gram-negative bacteria, adjusting the lipopolysaccharide (LPS), short chain fatty acid (SCFAs), pancreatic glucagon like peptide 1 (glp-1) and so on protein content, absorb medications affect blood glucose metabolism, polysaccharide absorption metabolism, to iridoids for the further study of mechanism of diabetes, and pay attention to the connection between various mechanisms.

### 3.5. Antitumor Activities

The antitumor activities of gentiopicroside in the non-small ovarian cancer cell line (SKOV3) have been reported. The antitumor effects were investigated by exposing the SKOV3 cells to gentiopicroside (0, 7.5, 15, 30 µM) for 24 h. Thereafter, anticancer effects were determined by MTT assay, the apoptosis was investigated by DAPI and annexin V/propidium iodide (PI) double staining, mitochondrial membrane potential (MMP) determination and cell cycle analysis were carried out by flow cytometry, and protein expression was examined by western blotting. The results showed that gentiopicroside exhibited anticancer effects on SKOV3 cancer cells in a dose-dependent manner. The IC50 of gentiopicroside was found to be 20 µM against the SKOV3 cancer cells. The anticancer effects were mainly found to be a result of the loss of MMP and the induction of apoptosis. The Bax/Bcl-2 expression ratio was also altered upon gentiopicroside treatment. Furthermore, gentiopicroside could arrest the SKOV3 cells in G2/M phase of the cell cycle and prevent their migration and invasion. These results indicated that gentiopicroside could prove as a potential lead molecule in the treatment and management of ovarian cancer [84].

Valjatrate E extracted from *Valeriana jatamansi* Jones is a kind of iridoid with reported anti-tumor activity. The effect of HepG2 on tumor invasion, on metastasis of human hepatocellular carcinoma and its potential mechanisms have been reported. The effects of valjatrate E on the migration and invasion of HepG2 cells were determined by the wound healing experiment and the transwell room experiment, respectively. The adhesion properties of HepG2 cells were evaluated by homogeneous and heterogeneous adhesion experiments. Finally, the molecular mechanism of valjatrate E inhibiting invasion and migration of HepG2 cells was studied by gelatine enzyme spectrometry and western blot. The valjatrate E was found to exhibit antitumor properties by inhibiting metalloprotease 2 (MMP-2) and metalloprotease 9 (MMP-9) expression; by inhibiting heterogeneous adhesion; by blocking mitogen-activated protein kinase (MAPK) signal; and by inhibiting the phosphorylation of extracellular signal-regulating kinase (p-ERK). These findings provided new evidence for the important role of the MAPK/ERK signaling pathway in promoting invasion and metastasis of HepG2 cells through p-ERK. The MAPK/ERK signaling pathway may, therefore, be a therapeutic target for tumors [85].

A novel secoiridoid compound, swertiamarin B, isolated from the ethanol extract of the aerial parts of *Swertia mussotii* has been shown to possess antitumor activity, which has been reported against four human tumor cell lines (HCT-116, HepG2, MGC-803 and A549). The results showed that swertiamarin B had strong cytotoxic activity against MGC-803 cell line (IC50 = 3.61) and could effectively inhibit cell proliferation, suggesting that it may be a new therapeutic agent for gastric cancer [86]. The research of the anti-tumor of natural medicine has been the hotspot of scientists. Although it has been proved that iridoid compounds have antitumor activity, most of them are based on in vitro cell experiments, but few in vivo and clinical research, and the effects of the compound in vivo and in vitro may be different. It is necessary to pay attention to the combination of in vivo and in vitro experiments in the future.

### 3.6. Others

Apart from the actions mentioned above, iridoids also exhibit other effects, including antioxidant [44], cardiocerebrovascular activity, cholagogic effects, and α-glucosidase inhibitory activity. Table 2 summarizes the other activities of iridoids.

## 4. Pharmacokinetic of Iridoids

In recent years, iridoids have been proven to have a wide range of biological activities, and their pharmacokinetic characteristics have also been widely studied [112]. The different structures of parent nuclei of iridoids result in different metabolic pathways, mainly through the hydrolysis of β-glucosidase, glucose-aldeacidification, isomerization, and other metabolic processes throughout the body [8]. The pharmacokinetics have been defined based on the different structures of the parent nucleus.

### 4.1. Iridoid Glycosides

The pharmacokinetics of iridoid glycosides are summarized based on several active compounds as examples, including geniposide and agnuside.

In general, glycosides do not work directly in the blood, but in the digestive system enzymes, stomach acid, intestinal bacteria, intestinal membrane enzymes, and other changes into another structure, and then plays a role. The in vivo and in vitro studies using the intestinal perfusion and Caco-2 model demonstrated that geniposide absorbed by passive diffusion had better absorption in the duodenum and jejunum from the intestinal perfusion model, suggests that the correlation between the pattern of use and bioavailability. On the other hand, verapamil influenced the transportation of geniposide, while EDTA did not which indicated that the absorption of geniposide in vivo may involve the active efflux mechanism mediated by P-glycoprotein [113]. The discovery of gardenoside in the brain suggests that it can pass through the blood-brain barrier for therapeutic purposes [114]. The pharmacokinetics of geniposide in rats with type 2 diabetic rats had higher C_max_, larger area under curve (AUC), longer T_max_, lower clearance rate (CL), and shorter mean residence time (MRT) (*p* < 0.05), compared to the healthy rats. The variable results that are attributed to geniposide can be hydrolyzed to aglycone by β-glucosidase produced by intestinal epithelia cells and absorbed into the blood circulation in the type 2 diabetic rats [115]. Yang et al. studied the effects of *G. jasminoides* extract on rat liver, and the oral results showed that the extract was significantly better than genipin, possibly because the main component of the water extract, gardenia glycoside, was better dispersed in the gastrointestinal tract and was easier to be absorbed through intestinal epithelial cells after being decomposed into ginipin by intestinal microorganisms [116]. The metabolic processes of genistein are shown in Figure 5. After intravenous administration of naosuning injection mainly containing geniposide, 70 percent of the urine was excreted within 10 h, suggesting that excretion through the kidney was the main excretion mode of the geniposide [117]. The metabolic process of geniposide is shown in Figure 7.

Similarly, one study determined the content of agnuside in mouse plasma and various tissues such as liver, spleen, kidney, lung, and brain. The total agnuside exposure (AUC_0→∞_) was highest in the small intestine, followed by kidney and liver, while the heart had the lowest of it. AGN is detected in the brain within 24 h, indicating it could effectively cross the blood-brain barrier and the selective retention in the brain. When taken orally for 0.5 h (T_max_), the serum concentration of agnuside increased rapidly (C_max_ = 422.33 ± 10.73 ng/mL, half-life (T_1/2_) = 0.90 ± 0.16 h), indicating that agnuside could be rapidly absorbed and distributed in tissues. At the same time, the apparent distribution volume (Vd) of agnuside was 143.02 ± 79.02 L/kg, indicating that agnuside is more widely distributed in tissues than in plasma. On the other hand, the C_max_ value of agnuside when injected intravenously (62,852.32 ± 16,013.94 ng/mL) was significantly higher than that of the oral administration. It is the highest in the intestine, which may be the reason for its low oral bioavailability, and it could be cleared out of all tissues within 8 h. Agnuside had the highest content in the intestine, followed by the kidney, indicated that it plays an important role in the extrahepatic clearance as the main excretory organ [118]. At present, there have been some studies on the pharmacokinetics of iridoids, but only limited to a few compounds, the pharmacokinetics of this class of compounds should be fully explored, and their metabolites and mechanisms should be comprehensively and systematically studied.

### 4.2. Secoiridoids

The metabolism of the secoiridoids such as gentiopicroside, swertiamarin, sweroside, and picroside-I, II, and III has been reviewed in the following section.

The structure of both gentiopicrin and sweroside are similar to swertiamarin, but for swertiamarin, it is hard to penetrate the blood-brain barrier [119]. In the model of rats, gentiopicrin and swertiamarin are found to be hydrolyzed by bacterial beta-glucosidase to form aglycones, which is then transformed into iso-coumarin derivative erythrin lactone. Under the action of the liver and intestinal bacteria, gentiopicrin undergoes reactions such as isomerization and lysis to form the final metabolite to be expelled from the body, where swertiopicrin is converted into isocoumarin and nitrogen-containing metabolite [120]. The isocoumarins and alkaloids produced during metabolism are the pharmacodynamic components of gentiopicrin and Swertia picroside in rats [121,122]. Another study showed that gentiopicroside might be inhibited or induced by cytochrome P450 enzymes, which, which can contribute to the gentiopicroside metabolism [123,124,125]. In feces samples, it is speculated that GPS reportedly transforms into five active constituents, SWS is metabolized by intestinal bacteria by the first-pass effect. Gentiopicroside, swertiamarin, and sweroside are mainly excreted by the kidneys and in lower amounts in the bile and feces [126]. The metabolic processes of gentiopicrin are shown in Figure 8.

High-performance liquid chromatography-electrospray tandem mass spectrometry (LC-ESI-MS/MS) was used to determine the picroside I-III levels in rat plasma and tissue homogenates. Three compounds were found to be widely distributed and quickly eliminated. In addition, picroside I-II were also found to be able to rapidly pass the blood-testicular barrier, but not the blood-brain barrier [127]. In addition, the highest concentrations of picroside I and III were found in kidneys, while picroside II was found in the liver. Moreover, it was found that picroside II exhibits the slowest metabolism with the terminal elimination half-lives (T1/2) were reported to be 0.941 ± 0.235 h [96]. Another study showed that because picroside I, II has a lipophilic and ionizable state, it can be completely metabolized [128].

### 4.3. Others

In addition to the aforementioned compounds, bis-iridoids and non-glycosidic iridoids have a wide range of pharmacological effects, and their in vivo metabolism has also been studied.

Studies have found that intestinal bacteria can convert geniposide to genipin. After sulfatase (type H-1) hydrolysis of plasma samples, many genipins appeared, indicating that genipin sulfate is the main metabolite of genipin. Furthermore, both oral administration and injection of genipin could quickly detect the presence of ginipin sulfate, indicating that genipin sulfate is produced when passing through the intestines and liver. When administered to rats at doses of 200 mg/kg, genipin was found to be highly toxic. Therefore, the dose of genipin should be properly formulated to ensure safety [129]. Another study found that the plasma concentration of apocynin can reach C_max_ at 504.42 ± 1.14 ng/mL within 6 h, and it has the largest AUC_0-t_ value compared with picroside I and II. Apocynin can be quickly absorbed but is eliminated slowly. Meanwhile, HPLC analysis of urine from the rat, showed no detection of apocynin, indicating that apocynin can be completely metabolized in vivo, which could be because of its high lipophilicity and ionizability [128].

Iridoids mostly exist in the form of glycosides, and its metabolic process has undergone β-glycosidase enzymes hydrolysis, glucose metabolism aldehyde acidification, and isomerization. However, in vivo, metabolism is a complex process, the metabolism of each compound has its own characteristics, the metabolic mechanism of the compound is not limited to a certain way. At present, the research on the pharmacokinetics of iridoids mainly focuses on the pharmacokinetics of medicine absorption, distribution, metabolism, and excretion, with less research on the mechanism and more focused on intestinal metabolism, but less on the mechanism of other metabolic sites in vivo. Therefore, we should systematically study the metabolic mechanism of iridoid and consider the interaction between the mechanisms.

## 5. Conclusions

Iridoids are a type of natural products widely distributed in the plant kingdom, which have a variety of biological activities and prodrug characteristics, with high medicinal value. Modern pharmacological studies have found that these compounds have anti-tumor, anti-depression, liver-protecting, neuroprotective, and other biological effects. The anti-tumor activity and therapeutic effects of iridoid on diabetes have been widely studied; it was found that ether terpenes can also inhibit DNA polymerase activity, suggesting that new anti-tumor medicine can be developed to inhibit DNA synthesis.

In recent years, with the continuous improvement in extraction methods and storage conditions, increasing numbers of iridoids have been identified. However, they are prone to degradation under both physical and chemical stress conditions and have poor chemical stability, which produces limitations in the study of their monomers, and hinders the study of their activities and functions. Moreover, most of the studies on iridoids are still focused on the determination of their structures and analysis of their activities. There is a lack of systematic research on their structure types and structure-activity relationship. Therefore, the analysis of and comparative studies on the structure and function of iridoids should be further intensified, to better confirm the main active functional groups and effects of iridoids, and provide effective data support for chemical modification and new medicine development, which will be a new hotspot of studies on natural products.

Several recent studies have shown that a variety of iridoids can be used as a therapeutic medicine for intracerebral targeting because they can pass the blood-brain barrier. However, the content distribution in the brain is low, and the compounds undergo rapid absorption and elimination in vivo and wide distribution to tissues and organs and have low biological benefits when administered by the oral route. The metabolic process for iridoids in vivo was mainly studied in animal models such as rats and mice; however, the metabolic process and pharmacokinetic parameters in the human body have been rarely studied. There are a few toxic and side effects reported—however, there is a lack of research on the preparation of iridoid monomers for use in clinical treatment. Therefore, in-depth studies on iridoids are necessary to improve their bioavailability and to develop new dosage forms after a detailed study on their toxicity.

In summary, recently, new structures and effects of iridoids have been elucidated at a rapid rate. In vivo characterization of iridoids will help determine the pharmacodynamic properties and explore the therapeutic potential of compounds used in medicine. The types, constituents, pharmacological activities, and metabolism of iridoids have been systematically summarized in this review to lay a foundation for further studies on iridoid glycosides.

## Figures and Tables

**Figure 1 molecules-25-00287-f001:**
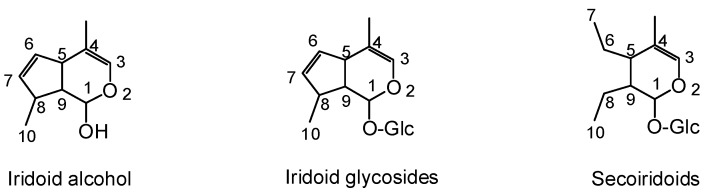
Three basic skeletons of iridoids.

**Figure 2 molecules-25-00287-f002:**
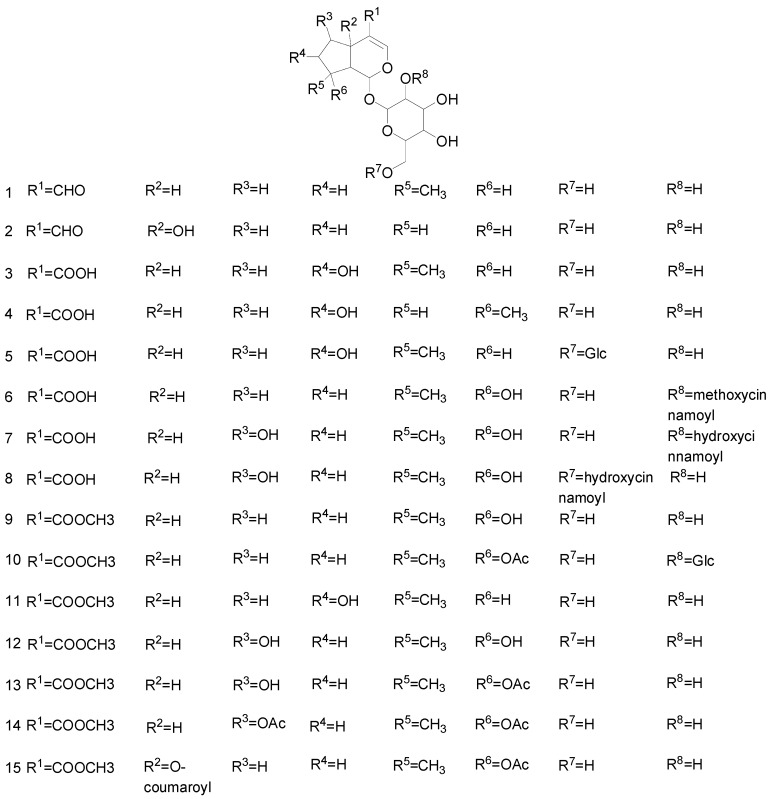
Chemical structures of iridoid glycosides.

**Figure 3 molecules-25-00287-f003:**
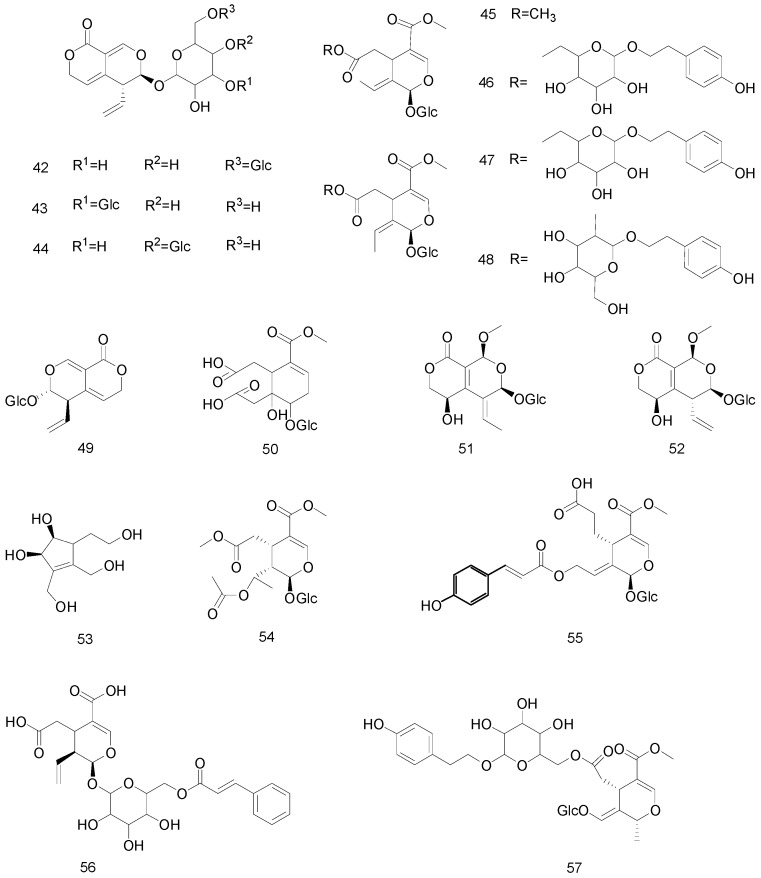
Chemical structures of secoiridoids.

**Figure 4 molecules-25-00287-f004:**
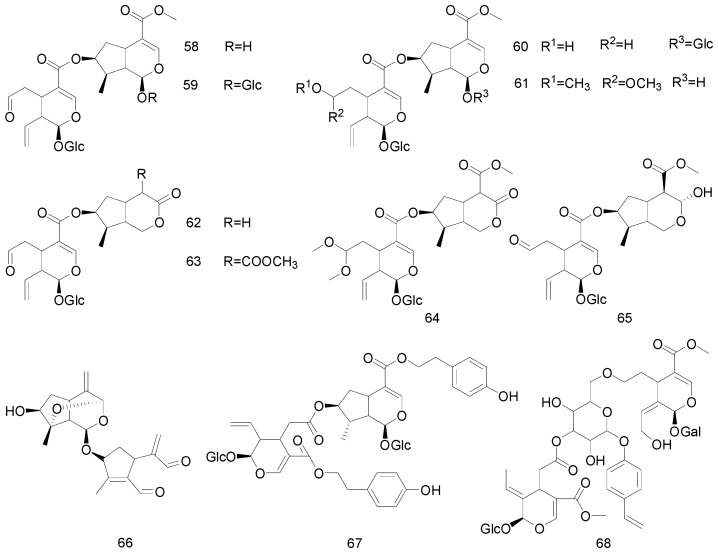
Chemical structures of bis-iridoids.

**Figure 5 molecules-25-00287-f005:**
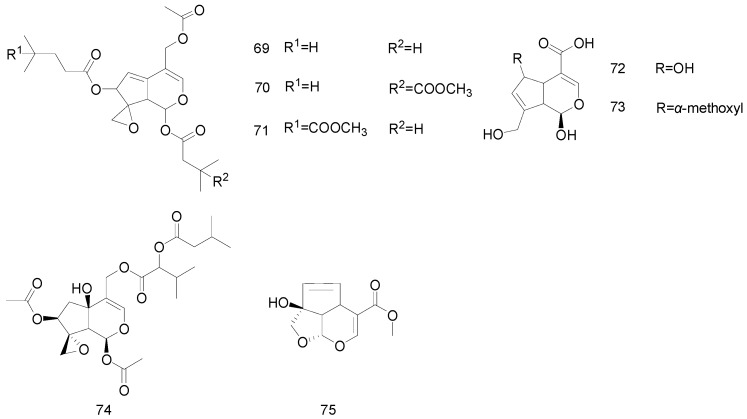
Chemical structures of non-glycosidic iridoids.

**Figure 6 molecules-25-00287-f006:**
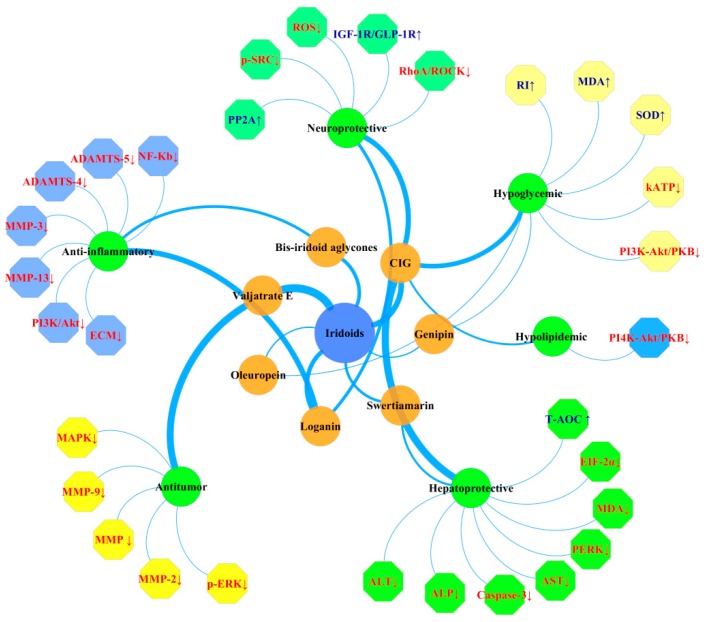
The pharmacological activity mechanism of iridoids. The blue arrow up indicates a positive or upregulated effect, while the red down arrow indicates a negative or downregulated effect. Orange circles represent some activity constituents, green ones represent the common pharmacological activities of iridoids, while represented enzymes and signaling pathways are illustrated by polygons. Abbreviations here represent the same meaning as in the body text.

**Figure 7 molecules-25-00287-f007:**
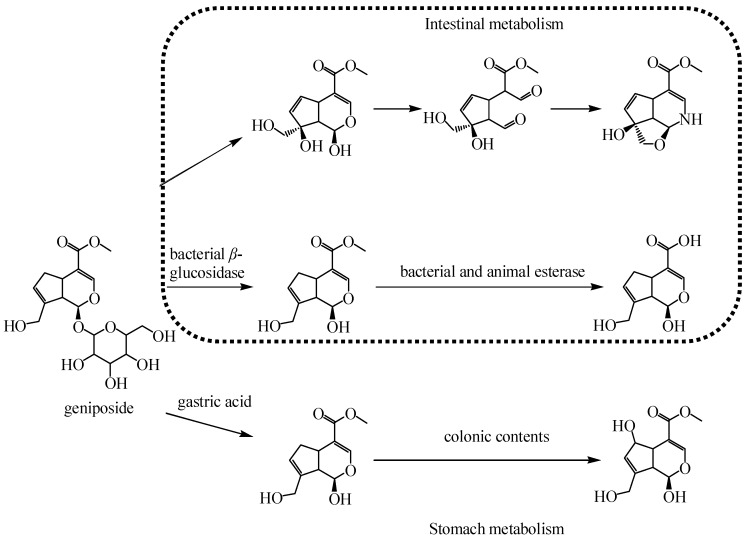
The metabolic processes of geniposide.

**Figure 8 molecules-25-00287-f008:**
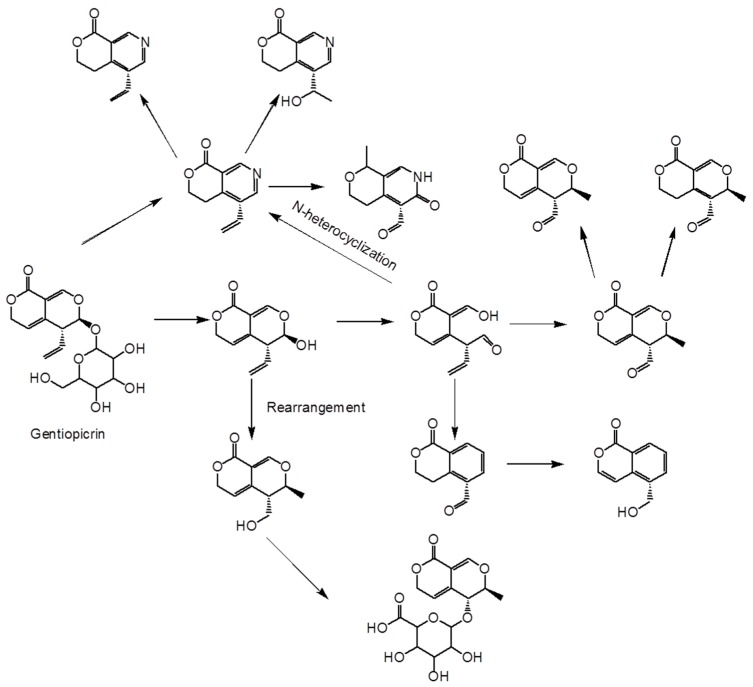
The metabolic processes of gentiopicrin.

**Table 1 molecules-25-00287-t001:** Active iridoids compounds from 2009 to 2019.

No.	Classes	Compounds	Sources	Ref.
1	Iridoid glycosides	Euphroside	Scrophulariaceae	[13]
2		Plantarenaloside	Plantaginaceae, Bignoniaceae	[14,15]
3		Geniposidic acid	Scrophulariaceae, Labiatae	[13]
4		Loganic acid	Acanthaceae	[16]
5		Loganic acid-6′-*O*-β-d-glucoside	Gentianaceae	[17]
6		2′-*O*-(4-methoxycinnamoyl) mussaenosidic acid	Acanthaceae	[18]
7		2′-*O*-*trans*-coumaroylshanzhiside	Rubiaceae	[19]
8		6′-*O*-*trans*-coumaroyl-shanzhiside	Rubiaceae	[19]
9		Mussaenoside	Scrophulariaceae	[13]
10		Lupulinoside	Acanthaceae	[20]
11		8-epideoxyloganic acid	Bignoniaceae	[14]
12		Shanzhiside methyl ester	Acanthaceae	[20]
13		Barlerin	Acanthaceae	[20]
14		Acetylbarlerin	Acanthaceae	[20]
15		6-*O*-*trans*-*p*-coumaroyl-8-*O*-acetylshanzhi side methyl ester	Acanthaceae	[20]
16		6′-*O*-acetylgeniposide	Rubiaceae	[21]
17		6′-*O*-*trans*-*p*-coumaroylgeniposide	Rubiaceae	[21]
18		6′-*O*-*trans*-*p*-coumaroyl-geniposidic acid	Rubiaceae	[21]
19		6′-*O*-*trans*-sinapoylgeniposide	Rubiaceae	[21]
20		Pinnatoside	Verbenaceae	[22]
21		6β-ethoxygeniposide	Rubiaceae	[19]
22		10-*O*-acetylgeniposide	Rubiaceae	[21]
23		Eurostoside	Scrophulariaceae	[23]
24		10-*O*-succinoylgeniposide	Rubiaceae	[21]
25		Ninpogenin	Scrophulariaceae	[24]
26		8-*O*-(coumaroyl)harpagide	Scrophulariaceae	[24]
27		6-*O*-α-d-galactopyranmosylharpagoside	Scrophulariaceae	[24]
28		8-*O*-feruloylharpagide	Scrophulariaceae	[24]
29		Dihydropenstemide	Plantaginaceae	[15]
30		Patrinalloside A	Valerianaceae	[25]
31		10-isovaleroyl-dihydropenstemide	Loganiaceae	[15]
32		Buddlejoside A9	Loganiaceae	[26]
33		6-*O*-α-L-(″-acetyl-4″-*O*-*trans*-isoferuloyl) rhamnopyranosyl catalpol	Loganiaceae	[26]
34		6-*O*-α-L-(4″-*O*-*trans*-cinnamoyl) rhamnopyranosyl catalpol	Loganiaceae	[26]
35		6-*O*-α-L-(4″-*O*-*trans*-*p*-coumaroyl) rham-nopyranosyl catalpol	Loganiaceae	[26]
36		Brasoside	Verbenaceae	[27]
37		Deacetyl asperuloside	Rubiaceae	[28]
38		Asperuloside	Rubiaceae, Plantaginaceae	[28,29,30]
39		8α-butylgardenoside B	Rubiaceae	[19]
40		Amphicoside	Scrophulariaceae	[31,32]
41		Specioside	Bignoniaceae	[33]
42	Secoiridoids	6′-*O*-β-d-glucopyranosyl gentiopicroside	Gentianaceae	[34]
43		3′-*O*-β-d-glucopyranosyl gentiopicroside	Gentianaceae	[34]
44		4′-*O*-β-d-glucopyranosyl gentiopicroside	Gentianaceae	[34]
45		Oleoside dimethyl ester	Oleaceae	[35]
46		(8*E*)-nuezhenide	Oleaceae	[35]
47		(8*Z*)-nuezhenide	Oleaceae	[35]
48		(8*Z*)-nuezhenide A	Oleaceae	[35]
49		Gentiopicroside	Gentianaceae	[17,34,36,37,38]
50		Oleuropein	Oleaceae	[35,39]
51		Qinjiaosides B	Gentianaceae	[34]
52		Qinjiaoside A	Gentianaceae	[34]
53		7-hydroxy eucommiol	Bignoniaceae	[33]
54		7-methoxydiderroside	Acanthaceae	[20]
55		Isojaslanceoside B	Oleaceae	[40]
56		6′-*O*-*trans*-cinnamoyl-secologanoside	Oleaceae	[40]
57		Nuezhenelenoliciside	Oleaceae	[40]
58	Bis-iridoids	Sylvestroside III	Dipsacaceae	[41]
59		Cantleyosid	Dipsacaceae	[41]
60		Sylvestroside I	Dipsacaceae	[41]
61		Sylvestroside IV Dimethyl acetal	Dipsacaceae	[41]
62		Laciniatoside II	Dipsacaceae	[41]
63		Sylvestroside IV	Dipsacaceae	[41]
64		sylvestroside III dimethyl acetal	Dipsacaceae	[41]
65		Laciniatoside I	Dipsacaceae	[41]
66		Polystachyn A	Valerianaceae	[42]
67		Sclerochitonoside C	Acanthaceae	[43]
68		Oleoside dimethyl ester	Oleaceae	[35]
69	Non-glycosidic iridoids	Valtrate	Valerianaceae	[44]
70		1-β-acevaltrate	Valerianaceae	[44]
71		Acevaltrate	Valerianaceae	[44]
72		6α-hydroxygenipi	Rubiaceae	[19]
73		6α-methoxygenipin	Rubiaceae	[19]
74		IIHD-acevaltrate	Valerianaceae	[42]
75		garjasmine	Rubiaceae	[19]

**Table 2 molecules-25-00287-t002:** Other activities of iridoids.

Function	Compounds	Dose	Model	Efficacy Evaluation	Ref.
Cardiocerebrovascular activity	Geniposide	33.2 μg/mL	Brain microvascular endothelial cell (BMEC) with oxygen–glucose deprivation (OGD)	Declining the productions of IL-8, IL-1β and monocyte chemotactic protein 1 (MCP-1)	[87]
Digestive activity	Gentiopicroside	20 mg/kg	Male rats (Sprague Dawley; six weeks old)	In all experimental groups an increase of gastric juice volume, total and free HCl concentration as well as pepsin concentration was observed.	[32]
Cholagogic effects	Genipin	1 μmol/100 g·min	Male sprague dawley rats and hyperbilirubinemic rats	Enhanced the ability of hepatocytes to secrete independent bile salts	[88]
Antioxidant	Geniposide	12.5, 25, 50 μg/mL	Human umbilical vein endothelial cell	Increased the activities of SOD, GSH-Px, NOS and NO production	[89]
Antithrombotic activity	Geniposide and genipin	20, 40 mg/kg	Male ICR mice	Significantly prolonging the time required for thrombotic occlusion	[90]
Anti-senescence	Catalpol	0.01, 0.1, 1 mg/mL	Human fibroblast Hs68 cell line irradiated by ultraviolet B (UVB)	Inhibited the formation of matrix metalloproteinase-1 (MMP-1)	[91]
Against intestinal ischemia/reperfusion (I/R) injury	Catalpol	25, 50 mg/kg	In vivo intestinal I/R-injured rats	Significantly attenuated rat intestinal I/R injury by decreasing pro-inflammatory cytokines, reducing oxidative stress, and restoring intestinal barrier function	[92]
Anti-depressive	Genipin	100 mg/kg	CUMS rat model	Decreases in serum trimetlylamine oxide (TMAO) and β-hydroxybutyric acid (β-HB)	[93]
Wound healing properties	Methylcatalpol	50 mg/kg	Male rabbits model	Protective activity against increased skin vascular permeability	[94]
Antiallergic	3,4-Dihydroxy catalpol	30 mg/kg	Asthmatic mouse model	Exhibited an antiasthmatic effect by the suppression of elevated IgE, IL-4 and IL-13 levels and eosinophilia in the plasma	[95]
Analgesic activity	Bis-iridoid	50, 100 mg/kg	Male mice	Bis-iridoid inhibited TNF-α-induced NF-κB-dependent promoter activity in a dose-dependent manner, The release of several proinflammatory cytokines and mediators (including TNF-a andPGE2) contributes to nociceptor sensitization and a reduction	[7]
Anti-HIV-1 activity	2′-*O*-(4-Methoxycinnamoyl) mussaenosidic acid	0.1 μg/mL	Real-time polymerase chain reaction (PCR) assay and HIV-1 p24 antigen kit	The expression level of C-C chemokine receptor type 5 (CCR5) and chemokine receptor type 4 (CXCR4) on CD4+ Tcellswere decreased in cells treated with this iridoid glycoside, demonstrated that this iridoid glycoside restricts HIV-1 replication on the early stage of HIV infection	[35]
Antiproliferative activities	Deacetyl asperuloside	806.4 μg/mL	K562 chronic myelogenous leukemic cells	Significantly increased caspase 3 activity (*p* < 0.05)	[16]
AChE inhibitory	Lupulinoside	134.0 μM	GST, AChE	Lupulinoside exhibited different levels of GST, AChE inhibitory	[41]
Anti-obesity	Genipin	20 mg/kg	High-fat diet–fed obese mice	Regulating miR-142a-5p/SREBP-1c axis, led to the inhibition of lipogenesis	[96]
Anti-osteoporosis activity	Aucubin	1, 2.5, 5 μM	MG63 cells	Improved osteoblast differentiation and enhanced the levels of BMP2 (bone morphogenetic proteins-2) in MG63 cells	[97]
	Nuezhenelenoliciside	-	MC3T3-E1 cells	Increased the proliferation of pre-osteoblast MC3T3-E1 cells, possessed anti-osteoporosis activity	[11]
Inhibition of gastric lesions	Genipin	50, 100 mg/kg (p.o.)	HCl/ethanol-induced gastric lesions rat	Increasing ROS and ROS-induced NAPDH-oxidase (NOX) production and enabling gastric cancer cells to start the tumor cell apoptosis process via Egr1/p21 signaling pathway	[98]
Anti- Alzheimer’s disease	Gardenoside	-	Fruit-fly Alzheimer’s disease model induced by human Abeta protein over-expression	Suppressed the expression of immune-related genes in the brain	[99]
Anti-angiogenic	Geniposide	25–100 μM	NIH3T3 cell line	Dose-dependently inhibiting the growth of the transformed N1H3T3 cell line	[100]
Inhibition of rheumatoid arthritis	Geniposide	50 μM	MH7A fibroblast-like synoviocytes in patients with rheumatoid arthritis	Inhibition of TNF-α-stimulated cell proliferation and activation of the Ras-Erk1/2 pathway via upregulating miRNA-124a expression	[101]
α-Glucosidase inhibitory activities	Aldosecologanin	0.5 mM	Lipopolysaccharide (LPS)-induced nitric oxide (NO) production and α-glucosidase reagent	Inhibiting α-glucosidase with IC50 values of 1.08 ± 0.70 and 0.62 ± 0.14 mM	[102]
Anti-anxiety properties	Geniposide	20, 40 mg/kg	Male ddY mice 4 weeks of age	Increased the social interaction time and demonstrated to exert an anxiolytic effect in a dose- dependent manner	[103]
Antiviral	6-*O*-*trans*-p-coumaroyl-8-*O*-acetylshanzhiside methyl ester	42.2 μg/mL	The viral CPE assay	Have potent in vitro activity against respiratory syncytial virus (EC50 2.46 μg/mL, IC50 42.2 μg/mL)	[104]
Angiogenic properties	Aucubin	10 mg/kg	Female C57BL/6 mice	Induced angiogenesis via vascular endothelial cell growth factor (VEGF)/Akt/endothelial nitric oxide synthase (eNOS) signaling pathway	[105]
Sedative effect	Total iridoids	0.3, 0.6, 0.9 g/kg	Male mice	Reduce the number of autonomous activities, prolong the sleep time of mice, and strengthen the hypnotic effect of pentobarbital sodium	[106]
Anticonvulsant	Total iridoids	1.0, 1.5, 2.0 g/kg	Pentetrazol-induced mice epilepsy model	Reduce the mortality of mice and prolong the latent period of convulsion	[106]
Improving irritable bowel syndrome	Total iridoids	0.3 mg/kg	Irritable bowel syndrome model rats	Regulating TPH1 and MAO-A factor, reducing 5-HT expression in serum and visceral sensitivity in rats with irritable bowel syndrome	[107]
Antibacterial activity	Phloyoside I, Phlomiol, Pulchelloside I	-	12 different strains	Exhibited from low to moderate levels (MIC = 0.05–0.50 mg/mL) of antibacterial activity	[108]
Antimalarial activity	Epoxygaertneroside, Methoxygaertneroside, Gaertneroside, Acetylgaertneroside, Gaertneric acid	1.3, 2.3, 4.3, 5.4, and 7.1 mg/mL	MT-4 cells	Displayed antiamoebic activity with IC50 values of 1.3, 2.3, 4.3, 5.4, and 7.1 mg/mL	[109]
DNA polymerase inhibitory	Catalpol, 8-*O*-acetylharpagide, Harpagide	47.8 mM	Adequate primer/template DNA and nucleotide analogues	Inhibition of tag DNA polymerase activity with an IC50 value of 47.8 mM	[110]
Anti-melanogenesis activity	9-epi-6α–methoxy geniposidic acid, Asperulosidic acid, Deacetyl asperulosidic acid, Scandoside methyl ester	100 mM	B16 melanoma cells induced by a-melanocyte-stimulating hormone (a-MSH)	Exhibited anti-melanogenesis activity with 40–50% reduction of melanin content at 100 mM	[111]

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
