# Peer review of "Iridoids: Research Advances in Their Phytochemistry, Biological Activities, and Pharmacokinetics"

_molecules, 2020, doi:10.3390/molecules25020287_

Round 1

Reviewer 1 Report

This is a comprehensive review on phytochemistry and pharmacology of iridoids.

Genipin is an important member and some its chemistry and pharmacology  may be extended a bit more.

The strength of this review is on an overall introduction to the class of iridoids and a summary of important examples on structures and biological activities, which seems to have  limited number of  reviews from different perspectives.   The weakness of this review is perhaps a lack of organization on describing the activities of so many bioactive Iridoids. The table 2 is a good attempt for a succinct summary. However, the pharmacology  part can be further improved by featuring on recent advances on several important compounds such as geniposide/genipin, catalpol, loganin and also some of the less known. A special account can be given on genipin, highlighting its crosslinking activity, impacts on mitochondria,  anti-diabetics , anti-tumour activities and applications in drug delivery. The pharmacokinetics part may be shortened with diagram/Table  showing the metabolites and key data.    minor: 
p.7, The sentence " laxoside listed in an international catalog does not describe its analogous  structure" is ambiguous. Please revise.

Author Response

Response to Reviewer 1 Comments

Dear Reviewer,

Thank you very much for your help with our paper (Manuscript ID: molecules-666146). We have revised our manuscript carefully based on each point raised in the review process. As suggested, the whole manuscript was improved. For your convenience, we have enclosed our revised manuscript where all changes remain marked in red.

The Responses to the reviewers' comments are as flowing:

Point 1: The weakness of this review is perhaps a lack of organization on describing the activities of so many bioactive Iridoids. The table 2 is a good attempt for a succinct summary. However, the pharmacology part can be further improved by featuring on recent advances on several important compounds such as geniposide/genipin, catalpol, loganin and also some of the less known. 

Response 1: Thanks a lot for the comments, which are very helpful for us to improve the manuscript. According to your suggestion, through further screening of the literature, we have summarized the pharmacological activity of the article and supplemented it. Please refer to page 13-16 in Table 2.

Point 2: A special account can be given on genipin, highlighting its crosslinking activity, impacts on mitochondria, anti-diabetics, anti-tumour activities and applications in drug delivery.

Response 2: Thanks a lot for the comments, this is a good idea to use graphics to express the content of the article. So, in the pharmacokinetics part, we added pictures instead of words to explain the metabolic mechanism of compounds in vivo. Please refer to page 18 and 19 Figure 7 and 8.

Point 3: The sentence "laxoside listed in an international catalog does not describe its analogous structure" is ambiguous.

Response 3: Thanks a lot for the comments. The sentence mentioned above means structures similar to those of laxoside have not been previously described in international catalogues. In addition, we have also modified the expression in the text. Please refer to page 7 line115-116.

Reviewer 2 Report

The authors claim that the aim of this manuscript is to provide a review of phytochemistry, biological activities, and pharmacokinetics of iridoids in the time spam 2009-2019. A search performed in ISI web of Science in that time span and searching for iridoids provide 984 references, a manual search witting the results for relevant results for 2019-2018 result in 102 references. The authors present 99 references for all time span. So, the reviewer think that this review cannot cover all the relevant literature to reach the goal.

Line 32  the plant kingdom      change to    the kingdom Plantae

in the dicotyledonous plants        The nomenclature, now, is monocotyledonous and eudicotyledonous so change in accordance.

Line 35-36   Since alcoholic hydroxyl groups belong to hemiacetal hydroxyl groups and are active in nature. Besides, they are mostly in the form of glycosides, a…       Something is missing where, this don’t make sense.

Line 48   Iridoids belong to monoterpenoids, which are acetal derivatives of antinodal.  ???? Add reference so it will be possible to understand what the authors are trying to say.

Line 49-50   Based on 50 whether it has a complete cyclopentane structure unit, it can be divided into two kinds:… Just simplify this phrase so it will be more understandable. Add a Figure with the 3 basic skeletons of iridoids.

Line 68 reported for the first time in Sambucus williamsii Hance [46].  Plant names must be put in italic, correct in accordance.

Line 72-75  The 1H detected heteronuclear multiple bond correlation (HMBC) spectrum of 10-O-β-D-glucosylborreriagenin was analyzed and found the correlation between H-10 with C-7, C-8 and with the anomeric carbon C-10 and of the anomeric proton H-10 with C-10 unambiguously establish a covalent ether linkage between the glucose moiety and C-10.  The aim of this manuscript is “phytochemistry, biological activities, and pharmacokinetics of iridoids” so this phrase is of no interest to the subject unless the elucidation of all structures are described, or this will lead to some kind of new activity, pharmacokinetics etc.

Line 83  At present, compound 45-61 are the active compounds of secoiridoid class…    Figure 2 of secoiridoids present compounds 42-57 so there is a discrepancy, correct.

Line 98-101 In addition to the above-mentioned compounds, there exists a class of bis-iridoids in nature, which are dimeric iridoid formed by the structure of two iridoids. The C-7 of an iridoid is linked to the C-4 of a secoiridoid to form a dicyclic ethers terpene. These compounds have been shown to have anti-inflammatory and analgesic activities.  Add references.

Line 311-312  Iridoids are natural secondary metabolites of plants, which have two basic skeletons of iridoid and secoiridoid, and most of them exist in the form of glycosides [8].   Just delete, already point it out.

Line 366  4.2 Pharmacokinetics of Secoiridoids   change to   4.2 Secoiridoids

Author Response

Response to Reviewer 2 Comments

Dear Reviewer,

Thank you very much for your help with our paper (Manuscript ID: molecules-666146). We have revised our manuscript carefully based on each point raised in the review process. As suggested, the whole manuscript was improved. For your convenience, we have enclosed our revised manuscript where all changes remain marked in red.

The Responses to the reviewers' comments are as flowing:

Point 1: The authors claim that the aim of this manuscript is to provide a review of phytochemistry, biological activities, and pharmacokinetics of iridoids in the time spam 2009-2019. A search performed in ISI web of Science in that time span and searching for iridoids provide 984 references, a manual search witting the results for relevant results for 2019-2018 result in 102 references. The authors present 99 references for all time span. So, the reviewer think that this review cannot cover all the relevant literature to reach the goal. 

Response 1: Thanks a lot for the comments. From 2009 to 2019, there were a lot of literatures on terpenoids of iridoids. We searched the literatures on websites such as Google Scholar, PubMed, CNKI and Baidu Scholar et al., and screened the literatures by searching the keywords “iridoids”, “iridoids phytochemistry”, “iridoids pharmacological”, “iridoids pharmacokinetic” et al. After targeted screening, we finally screened the literatures that can be used as evidence for the paper as references. Specific search methods and keywords in the text of the detailed description. Please refer to page 2 line 47-53.

Point 2: The plant kingdom change to the kingdom Plantae.

Response 2: Thank you for your comments, which are very helpful to my article. According to your suggestion, I have changed “plant kingdom” to “kingdom Plantae”. Please refer to page 1 line 32.

Point 3: Since alcoholic hydroxyl groups belong to hemiacetal hydroxyl groups and are active in nature. Besides, they are mostly in the form of glycosides, a… Something is missing where, this don’t make sense.

Response 3: Thanks a lot for the comments. This expression is really inappropriate, according to your opinion, we have changed it to Iridoids have hemiacetal hydroxyl group, and are active in nature. Besides, they are mostly in the form of glycosides, and are combined with glucose at the C-1 hydroxyl group. Please refer to page 1 line 35-36.

Point 4: Iridoids belong to monoterpenoids, which are acetal derivatives of antinodal. ???? Add reference so it will be possible to understand what the authors are trying to say.

Response 4: Thanks a lot for the comments. The meaning of this sentence is iridoids belong to monoterpenoids, and they are acetal derivatives of iridodial. In addition, we have added reference. Please refer to page 2 line 55.

Point 5: Based on 50 whether it has a complete cyclopentane structure unit, it can be divided into two kinds:… Just simplify this phrase so it will be more understandable.

Response 5: Thanks a lot for the comments. According to your opinion, we have changed it  to according to the integrity of the cyclopentane unit, it can be divided into two types. Please refer to page 2 line 56-57.

Point 6: Add a Figure with the 3 basic skeletons of iridoids.

Response 6: Thanks a lot for the comments. According to your suggestion, we have add a figure with the 3 basic skeletons of iridoids. Please refer to page 2 figure 1.

Point 7: Reported for the first time in Sambucus williamsii Hance [46]. Plant names must be put in italic, correct in accordance.

Response 7: Thanks a lot for the comments. According to your suggestion, we have revamped all of the format of plant names to italics, and correct them in accordance. And the specific changes in the article are all shown in red font. Such as page 4 line 78, 83, 84.

Point 8: The 1H detected heteronuclear multiple bond correlation (HMBC) spectrum of 10-O-β-D-glucosylborreriagenin was analyzed and found the correlation between H-10 with C-7, C-8 and with the anomeric carbon C-10 and of the anomeric proton H-10 with C-10 unambiguously establish a covalent ether linkage between the glucose moiety and C-10.  The aim of this manuscript is “phytochemistry, biological activities, and pharmacokinetics of iridoids” so this phrase is of no interest to the subject unless the elucidation of all structures are described, or this will lead to some kind of new activity, pharmacokinetics etc.

Response 8: Thank you for your comments, which will be of great help to my article. We agree with your suggestion that this sentence is inappropriate. We have deleted it and added new content. Please refer to page 4 line 83-85.

Point 9: At present, compound 45-61 are the active compounds of secoiridoid class…Figure 2 of secoiridoids present compounds 42-57 so there is a discrepancy, correct.

Response 9: Thank you for your comments. According to your suggestion, we have modified the serial number of the compound to 42-57 to ensure that the text part is consistent with the serial number of the picture. Please refer to page 6 line 91.

Point 10: In addition to the above-mentioned compounds, there exists a class of bis-iridoids in nature, which are dimeric iridoid formed by the structure of two iridoids. The C-7 of an iridoid is linked to the C-4 of a secoiridoid to form a dicyclic ethers terpene. These compounds have been shown to have anti-inflammatory and analgesic activities. Add references.

Response 10: Thank you for your comments. The expression of this sentence is not clear enough. We have modified this sentence to in addition to the aforementioned compounds, there exists a class of bis-iridoids in nature that are dimeric iridoids formed from the structures of two iridoids. Cantleyoside, laciniatoside I-II, sylvestroside I, sylvestroside III-IV, sylvestroside III dimethyl acetal, and sylvestroside IV dimethyl acetal are mainly bis-iridoids isolated from Pterocephalus hookeri (C.B. Clarke) Höeck, which have been shown to have anti-inflammatory and analgesic activities, and added references in the article. Please refer to page 7 line 106-110.

Point 11: Iridoids are natural secondary metabolites of plants, which have two basic skeletons of iridoid and secoiridoid, and most of them exist in the form of glycosides [8]. Just delete, already point it out.

Response 11: Thank you for your comments. According to your suggestion, In the Pharmacokinetic of iridoids section, we have deleted the duplicate statement. Please refer to page 17 line 359.

Point 12: 4.2 Pharmacokinetics of Secoiridoids change to 4.2 Secoiridoids

Response 12: Thank you for your comments. According to your suggestion, we have changed “Pharmacokinetics of Secoiridoids” to “Secoiridoids”. Please refer to page 18 line 409.

Reviewer 3 Report

Authors revised in this paper the recent advances in Iridoids phytochemistry, biological  activities, and pharmacokinetics. Since there aren’t recent specific review on the topic in my opinion it deserve to be published. Nevertheless some modification are needed in my opinion.

In the “INTRODUCTION” section, it would be interesting to understand the database used, and the criteria for manuscript selection

It seems that compound 2 in Table 1 is the same of compound 3

Revise the correctness of the chemical name of the compounds e.g. line 63 6-O-α-L-(2″-acetyl-4″-O-trans-isoferuloyl) should be 6-O-α-L-(2″-acetyl-4″-O-trans-isoferuloyl)

In section 2.4 Non-glycosidic iridoids, underline the origin and the chemical elucidation beside their bioactivity that should be removed and cited in the sequent section.

Revise along the text ul or ml, that should be uL or mL

It would be nice to have a picture resuming the activities, maybe the most recurrent molecular mechanisms of which iridoids are responsible

In the section 4. Pharmacokinetic of iridoids, it should be introduced natural compound Pharmacokinetics and not again what iridoids are

In vivo and in vitro should be in italics

A critic comment of each section is needed, and is mostly missing

Author Response

Response to Reviewer 3 Comments

Dear Reviewer,

Thank you very much for your help with our paper (Manuscript ID: molecules-666146). We have revised our manuscript carefully based on each point raised in the review process. As suggested, the whole manuscript was improved. For your convenience, we have enclosed our revised manuscript where all changes remain marked in red.

The Responses to the reviewers' comments are as flowing:

Point 1: In the “INTRODUCTION” section, it would be interesting to understand the database used, and the criteria for manuscript selection 

Response 1: Thank you for your comment. According to your opinion, it will be very meaningful. In the “Introduction” section, we have added the database used, and the criteria for manuscript selection. Please refer to page 2 line 47-53.

Point 2: It seems that compound 2 in Table 1 is the same of compound 3

Response 2: Thank you for your comments. According to your opinion, we have changed compound 3 to geniposidic acid in table 1. Please refer to page 2 table 1 compound 3.

Point 3: Revise the correctness of the chemical name of the compounds e.g. 6-O-α-L-(2″-acetyl-4″-O-trans-isoferuloyl) should be 6-O-α-L-(2″-acetyl-4″-O-trans-isoferuloyl).

Response 3: Thanks a lot for the comments. According to your opinion, we have revised the correctness of the chemical names of all the compounds. Specific modifications have been marked in red letters in the article. Such as page 2 compound 6, 7, 8.

Point 4: In section 2.4 Non-glycosidic iridoids, underline the origin and the chemical elucidation beside their bioactivity that should be removed and cited in the sequent section.

Response 4: Thanks a lot for the comments. According to your opinion, we have removed their bioactivity and underline the origin and the chemical elucidation. In addition, their bioactivities have been cited in the sequent section. Please refer to page 8 line 124-128, and page 13 line 351.

Point 5: Revise along the text ul or ml, that should be uL or mL

Response 5: Thanks a lot for the comments. According to your opinion, We've changed all the words " ul or ml " in the text to " uL or mL ". The details of the changes are marked in red. Such as Table 2 line 1,4.

Point 6: It would be nice to have a picture resuming the activities, maybe the most recurrent molecular mechanisms of which iridoids are responsible

Response 6: Thank you for your advice. According to your opinion, this is a great way to express compound metabolism. According to your opinion, We have added to the usual active mechanism map of iridoids. Please refer to page 18 figure 7 and page 19 figure 8.

Point 7: In the section 4. Pharmacokinetic of iridoids, it should be introduced natural compound Pharmacokinetics and not again what iridoids are.

Response 7: Thanks a lot for the comments. According to your suggestion, in the fourth part, we modified the paper to remove the redundant part of introducing iridoids, and sorted out the pharmacokinetics of them. Please refer to the fourth part.

Point 8: In vivo and in vitro should be in italics

Response 8: Thank you for your comments, which will be of great help to my article. According to your opinion, we have changed the font format of all " in vivo” and “in vitro " in this article to italic. The details of the changes are marked in red. Such as page 11 line 212, page 13 line 346, 347.

Point 9: A critic comment of each section is needed, and is mostly missing

Response 9: Thank you for your comments. According to your suggestion, We added comments to the article. The details of the changes are marked in red. Such as page 1 line 41-42, page 10 line 180-184.

Round 2

Reviewer 2 Report

Figure 6 is a good idea but at the present form is to small and only can be understand with a size of 300%. Letters and arrows are to small.

Reviewer 3 Report

Authors correctly addressed all suggestions.